# Genomic Sequence Analysis of the Multidrug-Resistance Region of Avian *Salmonella enterica* serovar Indiana Strain MHYL

**DOI:** 10.3390/microorganisms7080248

**Published:** 2019-08-09

**Authors:** Yan Lu, Yanjia Wen, Ge Hu, Yuqi Liu, Ross C. Beier, Xiaolin Hou

**Affiliations:** 1National Demonstration Center for Animal Experimental Education, Beijing University of Agriculture, Beijing 102206, China; 2Beijing Key Laboratory of Traditional Chinese Veterinary Medicine, Beijing University of Agriculture, Beijing 102206, China; 3United States Department of Agriculture, Agricultural Research Service, Southern Plains Agricultural Research Center, Food and Feed Safety Research Unit, College Station, TX 77845–4988, USA

**Keywords:** IS*26* element, genome sequence analysis, MDR genes, MDR region molecular structure, mercury resistance, *Salmonella enterica* serovar Indiana, Tn*21* structure, type I integrons

## Abstract

A series of human and animal diseases that are caused by *Salmonella* infections pose a serious threat to human health and huge economic losses to the livestock industry. We found antibiotic resistance (AR) genes in the genome of 133 strains of *S.* Indiana from a poultry production site in Shandong Province, China. *Salmonella enterica* subsp. *enterica* serovar Indiana strain MHYL had multidrug-resistance (MDR) genes on its genome. Southern blot analysis was used to locate genes on the genomic DNA. High-throughput sequencing technology was used to determine the gene sequence of the MHYL genome. Areas containing MDR genes were mapped based on the results of gene annotation. The AR genes *bla*_TEM_, *strA*, *tetA*, and *aac*(6′)-*Ib-cr* were found on the MHYL genome. The resistance genes were located in two separate MDR regions, RR1 and RR2, containing type I integrons, and Tn*7* transposons and multiple IS*26* complex transposons with transposable functions. Portions of the MDR regions were determined to be highly homologous to the structure of plasmid pAKU_1 in *S. enterica* serovar Paratyphi A (accession number: AM412236), SGI11 in *S. enterica* serovar Typhimurium (accession number: KM023773), and plasmid pS414 in *S.* Indiana (accession No.: KC237285).

## 1. Introduction

*Salmonella* species are a major worldwide bacterial cause of acute gastroenteritis [1,2]. Salmonellosis is commonly caused by non-typhoidal *Salmonella enterica* serotypes [2]. The Centers for Disease Control and Prevention (CDC) has estimated that in the United States, foodborne *Salmonella* causes about 1 million illnesses, 19,000 hospitalizations, and 380 deaths each year [3]. Children under the age of 5, the elderly, and people with weakened immune systems are the most likely to have severe infections [4]. Recently, there has been a shift in the *Salmonella* serotypes found in poultry production, especially in diverse geographical regions of the world [2]. *Salmonella* enterica serovar Indiana was first reported in China during 1984 and is now commonly found in animals, animal processing facilities, food, and people [5]. *Salmonella* Indiana has become an important human foodborne pathogen found in poultry [2]. In recent years, there have been reports of human infection with diarrhea caused by multidrug-resistant *S.* Indiana in many provinces in China [6]. Antibiotics are commonly used for the treatment of bacteremia caused by *Salmonella* [3]. However, with the overuse of antibacterial drugs, drug-resistant strains with multidrug-resistance (MDR) phenotypes have appeared in large numbers, making the treatment of salmonellosis progressively more difficult [7]. Studies have shown that acquired resistance genes may be carried on the genomic DNA of *Salmonella* [8]. Through a study of multiple-chromosomal-type resistant *Salmonella*, it was determined that most of these resistance genes were concentrated in a small genomic island called SGI1 (*Salmonella* genomic island 1), and the area on the genome where these resistance genes are located is called the MDR region [9]. Due to the mobility of the resistance genes in SGI1, it is of great significance when studying the acquisition and transmission of bacterial MDR. The horizontal gene transfer of antimicrobial resistance (AMR) determinants located on plasmids was thought to be the main cause of the rapid proliferation and spread of drug resistance [10]. However, strains harboring SGI1 tend to rapidly disseminate and be more virulent [11]. It has been shown that SGI1 has the same general chromosomal location in different *Salmonella* serotypes, indicating that its insertion occurs through site-specific recombination [12]. Gene replacement in the integron structure is another way to contribute to the variability of the SGI1 antibiotic resistance (AR) gene cluster, which may result in stronger virulence and the further exchange of AR genes between other multidrug-resistant bacteria [12].

In the early stages of this study, we initially did drug testing and found MDR phenotypes in *Salmonella* and wanted to diagnose the role of drug-resistance genes in these bacteria. PCR was thought to be the best method to use but we were unable to determine whether the resistance genes were located on one or two fragments of the genome. Ultimately, the result led to sequencing the whole genome. This study carried out a comparative analysis of the gene sequences in the MDR regions of the genome of a *S. enterica* subsp. *enterica* serovar Indiana strain MHYL isolate and explored possible sources and routes for obtaining AR genes. The work here may help understand the emergence and spread of bacterial resistance and may be useful for guiding the prevention and management of clinically utilized drugs for the control of *S.* Indiana with the MDR phenotype.

## 2. Materials and Methods

### 2.1. Bacterial Strains and Drugs Tested

In this study, 133 strains of *S.* Indiana were isolated from a poultry food production site in Shandong Province, China. For each of the isolates, the broth microdilution method was used to determine the minimum inhibitory concentration (MIC) of 19 commonly used clinical antibacterial drugs: amikacin, amoxicillin, ampicillin, cefazolin, ceftiofur, chloramphenicol, danofloxacin, doxycycline, enrofloxacin, florfenicol, gentamicin, kanamycin, nalidixic acid, norfloxacin, polymyxin E, sulfamethoxazole, sulfisoxazole, tetracycline, and trimethoprim. The MIC values were determined according to methods of the American Clinical Laboratory Standards Committee (CLSI) [13]. The strain MHYL isolate that we selected had resistance to 17 drugs, and was only susceptible to amikacin and polymyxin E, and can be used as a representative strain for later studies. *Escherichia coli* ATCC 25922 was used as a quality control for antimicrobial susceptibility testing [14].

### 2.2. Kits and Reagents

The bacterial genomic DNA extraction kit, DNA MarkerII, was purchased from Tiangen Biotechnology Co., Ltd., Beijing, China; the 2×Trans Taq-T PCR Super Mix was purchased from the Beijing Quanjin Biotechnology Co., Ltd., Beijing, China; the lambda DNA Marker was purchased from the Beijing Dingguo Changsheng Biotech Co., Ltd., Beijing, China; the DIG High Prime DNA Labeling Detection Starter Kit was purchased from Sigma-Aldrich Chemie Gmbh, Munich, Germany; and the HindIII endonuclease, and the 10× NEB Buffer was purchased from New England Biolabs, Ipswich, MA, USA.

### 2.3. Concentration and Purification

The bacterial genomic DNA was extracted according to the DNA Marker II bacterial genomic DNA extraction kit instructions. The genomic DNA was then digested with the HindIII endonuclease.

To produce the sample size needed for electrophoresis and Southern blot hybridization, the enzyme digestion DNA gel electrophoresis loading conditions and the optimization of the Southern blot hybridization method were carried out. The digested genomic DNA was reduced in volume using phenol-chloroform-isoamyl alcohol and ethanol precipitation.

### 2.4. Amplification of Drug-Resistance Genes and Southern Blot Restriction Mapping

Amplification of the β-lactam resistance gene *bla*_TEM_, aminoglycoside resistance gene *strA*, tetracycline drug-resistance gene *tetA*, amido alcohol drug-resistance gene *floR*, and the quinolone-resistance gene *aac(6′)-Ib-cr* were carried out. The primer sequences are shown in Table 1, and the PCR reaction parameters are shown in Table 2 [15].

Specific probes for each drug-resistance gene were prepared, and the primers were synthesized by the Beijing Dingguo Changsheng Biotechnology Co., Ltd., Beijing, China. Southern blot experiments were performed according to the manufacturer’s directions included with the DIG High Prime DNA Labeling Detection Starter Kit.

### 2.5. Sequence Analysis of the MDR Region of the Strain MHYL Genome

The avian *S.* Indiana strain MHYL was sent to the Beijing Baimaike Biotechnology Co., Ltd., Beijing, China, for genome-wide sequencing, assembly, and gene function annotation. Based on the results of gene function annotation, the structure map showing the drug-resistance gene regions on the strain MHYL genome was mapped using Vector NTI software (version 11.5.1; ThermoFisher Scientific, Waltham, MA, USA), which is a suite of sequence analysis and design tools.

### 2.6. Prediction of Strain MHYL Genome-Encoding Genes and Non-Coding Genes

The Glimmer software (version 3.02; Institute for Genomic Research, Rockville, MD, USA) was used to identify the coding sequences of the genome; the rRNA, tRNA, and miRNA of the sequenced strain; and the sequenced genes were compared with Clusters of Orthologous Groups (COGs), Kyoto Encyclopedia of Genes and Genomes (KEGG), Gene Ontology (GO), Swiss-Prot, and NR Protein databases. The data were subjected to BLAST alignment, and the gene function annotation was performed.

## 3. Results

### 3.1. Southern Blot Hybridization of Drug-Resistance Genes

Southern blot methodology was used to hybridize the resistance genes *bla*_TEM_, *strA*, *tetA*, *floR*, and *aac(6′)-Ib-cr* corresponding to the observed drug-resistance phenotypes before and after digestion of the strain MHYL genome. The extracted genomic DNA of strain MHYL remained intact as was shown using electrophoresis of the hybridization results before and after digestion of the MHYL genome, which had no plasmid DNA interference and was successfully treated with the HindIII endonuclease.

Specific probes for each drug-resistance gene were hybridized with the strain MHYL genome before and after digestion. The results showed that each drug-resistance gene was successfully hybridized and mapped to the strain MHYL genomic DNA both before and after digestion. The electrophoresis map of the strain MHYL genomic DNA and the hybridization results for each drug-resistance gene were compared, and the band sizes were consistent. The presence of each drug-resistance gene on the MHYL genomic DNA was determined. Subsequently, the positional relationship of the restriction fragments for each drug-resistance gene was compared and analyzed. The resistance genes from the strain MHYL genome were concentrated in three regions of different fragment sizes, 20 kb (*tetA* and *aac(6′)-Ib-cr*), 10 kb (*bla*_TEM_, *floR*, *aac(6′)-Ib-cr*), and 9.4 kb (*strA*).

### 3.2. Statistics of the Strain MHYL Genome Sequencing Data

Based on the data obtained using sequencing, a MHYL genome library was constructed. The genomic size of the sequenced strain was estimated to be 5.3 Mb, as determined using sequencing data and sequencing depth.

### 3.3. Screening and Clustering Statistics of Genes Related to Drug Resistance

The annotation information of the genome-encoding genes of the sequenced strains was screened, the genes related to drug resistance were sorted, and the drug-resistance genes were classified and counted as shown in Table 3. There are a variety of genes on the genome related to the observed drug-resistance phenotypes of the multidrug-resistant *S.* Indiana strain MHYL obtained from poultry. These genes encode for proteins that result in resistance to multiple drugs.

### 3.4. Assembly and Sequencing of the Strain MHYL Genome Gene Sequence

Since the genome-wide gene sequence of *S.* Indiana was not found in NCBI, the genomic gene sequence of *S.* Typhimurium was selected as a reference [16]. In the *S.* Indiana strain MHYL genome, MDR genes and mobile elements were mainly located in two non-adjacent regions of the genome, resistance region 1 (RR1) and resistance region 2 (RR2). The GC content of the whole MHYL genome was 51.62%, while the GC content of RR1 and RR2 was higher than that of the whole MHYL genome at 69.85% and 63.41%, respectively. The strain MHYL whole genome sequence data was uploaded to NCBI (SUBID: SUB805704; BioSample: SAMN03287656; BioProject: PRJNA273784).

### 3.5. Mapping of the MDR Region of the Strain MHYL Genome

The gene structure of the RR1 and RR2 sequences on the MHYL genome were mapped and are shown in Figure 1 and Figure 2. By comparing and analyzing the genetic structure of each drug-resistance area, the possible sources and modes of transmission of drug-resistance genes may be inferred.

As shown in Figure 1, the RR1 sequence contains genes conferring resistance to β-lactams, amide alcohols, aminoglycosides, sulfonamides, and heavy metals such as mercury and other non-metallic substances. According to the gene structure map, RR1 can be roughly divided into two parts: an upstream region and a downstream region.

In the upstream region of RR1, the Tn*21* skeleton structure was observed, and on the end was an incomplete IS*1* (*insAB*) fragment. The inverted repeat sequence IRTn*21* with a size of 9 bp was detected at both ends of the upstream region, and the integrase gene *int1* and trimethoprim resistance gene *dfrA7* were detected therein. The chloramphenicol resistance genes *cmlA* and *floR*, aminoglycoside nontransferable encoding gene *aadA*, quaternary ammonium compound resistance gene *qacE△1*, sulfonamide resistance gene *sul*II, aminoglycoside phosphotransferase encoding gene *aphA1*, carbapenemase encoding gene *nmcR*, chloramphenicol acetyltransferase encoding gene *catB4*, extended spectrum β-lactamases (ESBLs) encoding gene *bla*_OXA-1_, truncated *tniA* transposase, heavy metal mercury resistance gene *merP* and the mercury resistance operon *merEDACPTR*, and the IS*26* element was inserted therein to form a complete Tn*21* transposon structure. Among them, the *int1* gene and the IS*26* element constitute a complete integron structure, and the front-end of the *int1*-*dfrA7* gene had a 99% homology with the integron sequence of the *S.* Indiana carrying plasmid pS414 in GenBank (accession number: KC237285) [17]. A BLAST comparison of the upstream gene sequence of RR1 revealed that it had a structure similar to SGI11 (accession number: KM023773) in GenBank [18]. Both had incomplete IS*1* fragments, a Tn*21* skeleton structure, type I integrons, and resistance genes related to the heavy metal mercury. This structure is also like that found in the MDR plasmid IncHI1 of *S.* Paratyphi A [19]. The aminoglycoside phosphotransferase encoding gene *aphA1* was flanked by two isotropic or inverted IS*26* sequences, forming the IS*26*-*aphA1*-IS*26* structure. This structure was first discovered on an *Escherichia coli* plasmid p0111_1 reported in 2001 (GenBank registration, No.: AP010961). It was shown that the *aphA1* gene is inserted between two transposable IS*26* sequences to form part of the transposon Tn*5715* [20], which mediates the insertion and transposition of MDR genes in a variety of Gram-negative bacteria. The presence of this structure was detected in the plasmid and MDR region of the *Corynebacterium urealyticum* genome [21]. Compared with SGI11, there are four IS*26*-*aphA1*-IS*26* structures upstream of RR1, and the integrase *int1* gene is inserted between two IS*26*-*aphA*-IS*26* structures in forward and reverse repeats to form two IS*26*-*aphA*-IS*26*-*int1*-IS*26*-*aphA*-IS*26* structures. The resistance-related genes *floR*, *nmcR*, *catB4,* and *bla*_OXA-1_ were inserted within the two four-IS*26* repeat structures, and all the IS*26*s were preceded by a 14 bp inverted repeat (GGCACTGTTGCAAA), forming a set of transposons. The complex transposon structure, which integrates functions, may be conducive to capturing exogenous resistance genes.

There are many IS*26* mobile elements with reverse repeats at both ends in the downstream region of RR1. The bleomycin resistance gene *ble*, sulfonamide resistance genes *sul*I and *sul*II, aminoglycoside resistance gene *strB*, ESBL encoding gene *bla*_TEM_, and the aminoglycoside acetyltransferase encoding gene *aacC4* were inserted therein, and these mobile elements were part of a variety of composite transposons. It was suggested that the composite transposon structure consisting of two IS mobile elements with inverted repeats at both ends can mediate the movement of genes inserted between the IS elements [21]. Also, some IS mobile elements may cause the homologous recombination of the bacterial genome to occur, resulting in variability in the genetic structure of the MDR region [22].

As can be seen from Figure 2, RR2 contains the chloramphenicol acetyltransferase-encoding gene *cat*, IS*1* (*insAB*), IS*21* (*istAB*), the transposon Tn*7* encoding for at least three transposition genes (*tnsABC*), the gene cluster *pcoABCDR* providing tolerance to the metal element copper, and an incomplete Tn*1721*-like transposon encoding tetracycline resistance. The IS*1* and IS*21* gene sequences were compared and found to lack direct repeats at both ends. It has been shown that IS elements lacking direct repeats at both ends are more likely to be derived from unconventional recombination rather than through a transposition mechanism [23]. The copper metal tolerance efflux system encoding genes *pcoA*, *pcoB*, *pcoR,* and *pcoC* may be activated when Cu^2+^ is in excess, and the protein pcoC can bind to bacterial intracellular Cu^2+^ together with the proteins pcoA, pcoB, and pcoD is an inner membrane-spanning protein that is required for complete Cu^2+^ removal from the bacteria, thereby, exerting resistance to metallic copper [24,25]. In a study of the genomic gene sequence of *Acinetobacter baumannii*, a truncated MDR Tn*1721* transposon was found comprised of the tetracycline drug-resistance genes *tetA* and *tetR*, together with the *perM* gene and transposase gene *tnpA* [26]. The incomplete Tn*1721*-like transposon (accession number: CT025832) had 99% homology with the incomplete Tn*1721* transposon gene sequence obtained in this study with *S.* Indiana strain MHYL. By comparison analysis, multiple strains of *Salmonella* were found to have consistent genomes with strain MHYL.

## 4. Discussion

By comparing and analyzing the gene sequences of the two MDR regions on the MHYL genome, it was found that: (i) part of the MHYL genome sequence and the *S.* Paratyphi A strain carrying the plasmid pAKU_1 (accession number: AM412236) had the same skeleton structure, and with high homology; (ii) the type I integron skeletal structure of the MHYL genome had high homology with the plasmid pS414 (accession number: KC237285) carried by *S.* Indiana; (iii) the multiple repeating sequences with the IS*26* mobile element at both ends could mediate the integration of drug-resistance genes into the genome of a new strain through the transposition mechanism, and may also cause homologous recombination of the strain MHYL genome; (iv) two IS*26*-*aphA*-IS*26*-*int1*-IS*26*-*aphA*-IS*26* structures may be a novel complex integron structure, which is conducive to capturing foreign drug-resistance genes; (v) the IS*1* and IS*21* lacking direct repeat sequences found at both ends of RR2 may be derived from unconventional recombination; and (vi) *tetA* and *tetR* genes may be derived from the Tn*1721*-mediated transposition mechanism and be transmitted horizontally between strains. The results of the comprehensive comparative analysis indicated that multiple acquired resistance genes on the genome of the multidrug-resistant *S.* Indiana strain MHYL may be derived from homologous or non-homologous strain genomes or their resistance plasmids. Mediated by IS*26* elements and a variety of transposons, resistance genes may be integrated into the genome via transposition or homologous recombination by a stable propagation mechanism independent of drug selective pressure. At the same time, however, the exogenous genes on the genome of some strains may remain in the multiple transposon structures that mediate their movement. The MDR region on the MHYL genome is composed of IS*26* elements and is speculated to be highly transferable. With the help of transposons, there may be movement of resistance genes within the genome of the same strain and the horizontal transmission of resistance genes between genomes of different strains and plasmids, but the possibility of this mechanism requires further study.

On the other hand, it was found using BLAST analysis that the upstream gene sequence of RR1 and the SGI11 (accession number: KM023773) sequence carried on the genome of *S.* Typhi had the same skeleton structure with high homology, and the *yidC* gene was present downstream of RR1. It has been shown that SGI1 and its variants are usually located in the upstream region of the *yidC* gene [9]. Therefore, it is speculated that the upstream gene structure of RR1 is an amplified SGI11-like structure, and multiple integration and transposition elements exist on it. This structure may be more conducive to capturing exogenous resistance genes, thereby enhancing the tolerance of strains to various antimicrobial agents, effectively demonstrating MDR. Whether this structure is a new type of SGI1 subtype remains to be seen following further study.

## 5. Conclusions

In this study, a whole genomic gene sequence of avian MDR *Salmonella* serrata was obtained, which confirmed that the *S.* Indiana strain MHYL genome carries the exogenous AMR genes *bla*_TEM_*, strA, tetA, floR,* and *aac(6′)-Ib-cr* related to the drug-resistance phenotype. Also, the MHYL genome carried the heavy metal mercury resistance gene *merP* and the *mer* operon *merEDACPTR*. The *merR* gene is the universal regulator for the *mer* operon [27]. Either *merT* or *merP* is sufficient to specify a mercury transport system [28]. The *merA* gene encodes for a reductase that reduces Hg^2+^ to Hg^0^ [27]. The *tniA* transposase gene is usually present in integrons and composite transposons conferring antibiotic resistance [29]. The *S. enterica* Indiana strain MHYL genome carries genes conferring resistance to β-lactams, amide alcohols, tetracyclines, aminoglycosides, and sulfonamides. The drug-resistance genes are concentrated in two MDR regions, and there is an MDR region gene structure composed of type I integrons and with multiple complex transposons.

## Figures and Tables

**Figure 1 microorganisms-07-00248-f001:**
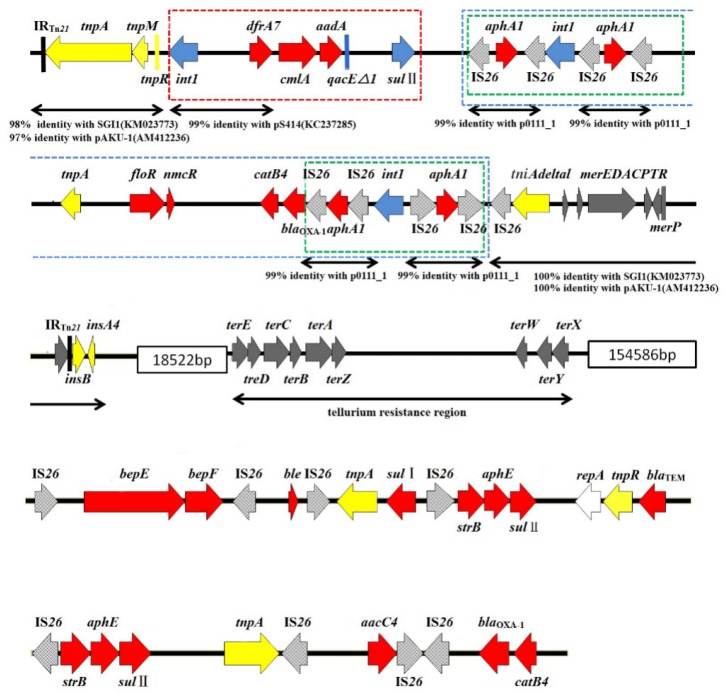
Schematic diagram of RR1. The red arrows in the figure represent drug-resistance gene cassettes, yellow arrows represent genes related to a transposition function, blue arrows represent the type I integrase genes and the integron-gene cassette 3′ conserved region gene structures, light gray arrows represent movable IS*26* elements, dark gray arrows represent heavy-metal- and non-metal-resistance genes, the red dashed box is a type I integron structure, the green dashed boxes are assumed to be a movable element structure, and the blue dashed boxes are assumed to be compound recombination. The structure of the arrow: the direction of the arrow shows the direction of gene coding, and the size of the arrow is drawn according to the size of the coding gene and the length of the gene sequence of the RR1 region.

**Figure 2 microorganisms-07-00248-f002:**
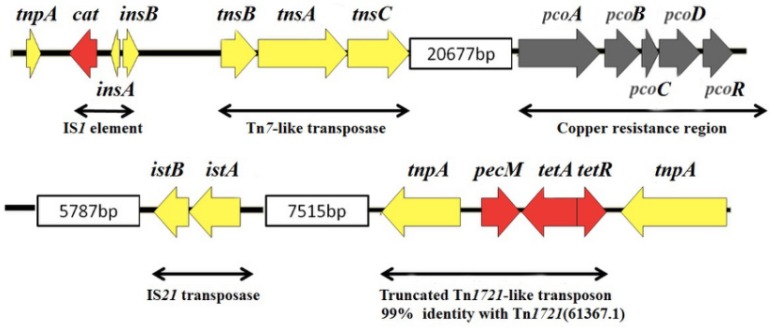
Schematic diagram of RR2. The red arrows in the figure represent the drug-resistance gene cassettes, the yellow arrows represent the genes associated with the transposition function, and the dark gray arrows represent the heavy metal resistance genes. The size of the arrow is in proportion to the size of the coding gene and the length of the RR2 gene sequence.

**Table 1 microorganisms-07-00248-t001:** Primer sequences of resistance genes.

Antibacterial Drug Type	Gene	Upstream primer (5′–3′) Downstream primer (5′–3′)	Registration No.	Fragment Size (bp)
β-lactam	*bla* _TEM_	CAGCGGTAAGATCCTTGAGA ACTCCCCGTCGTGTAGATAA	AY463797	643
Aminoglycosides	*strA*	CGACTTCTTACCGGACGAGGAC ACAGGTTGCGAAACGTGCCAAT	NC_009981	422
Tetracycline	*tetA*	GCTACATCCTGCTTGCCTTC CATAGATCGCCGTGAAGAGG	X75761	210
Amide alcohol	*floR*	TCCTGAACACGACGCCCGCTAT TCACCGCCAATGTCCCGACGAT	AJ251806	962
Quinolones	*aac(6’)-Ib-cr*	TTGCGATGCTCTATGAGTGGCTA CTCGAATGCCTGGCGTGTTT	EU543272	482

**Table 2 microorganisms-07-00248-t002:** PCR reaction parameters.

Gene	Initial Denaturation	Denaturation	Annealing	Extension	Cycles	Final Extension
*bla* _TEM_	95 °C, 4 min	94 °C, 30 s	48 °C, 30 s	72 °C, 1 min	30	72 °C, 10 min
*strA*	95 °C, 4 min	94 °C, 30 s	57 °C, 30 s	72 °C, 1 min	30	72 °C, 10 min
*tetA*	95 °C, 4 min	94 °C, 30 s	57 °C, 30 s	72 °C, 1 min	30	72 °C, 10 min
*floR*	95 °C, 4 min	94 °C, 1 min	63 °C, 1.5 min	72 °C, 1.5 min	30	72 °C, 10 min
*aac(6’)-Ib-cr*	95 °C, 4 min	94 °C, 45 s	55 °C, 45 s	72 °C, 45 s	30	72 °C, 10 min

**Table 3 microorganisms-07-00248-t003:** Classification of genes detected by PCR and BLAST on the strain MHYL genome.

Categories	Resistance Genes	Resistance Phenotypes
Beta lactams	*bla*_TEM_, *bla*_OXA_, *bla*_CTX-M-65_, *AmpC*, *nmcR*	Ampicillin, amoxicillin-clavulanic acid, cefazolin, ceftiofur
Aminoglycosides	*strA*, *aphA*, *aphE*, *strB*, *aacC4*, *aadA*, *aadA1*	Trimethoprim, gentamicin, kanamycin, amikacin
Tetracyclines	*tetA*, *tetR*	Tetracycline, doxycycline
Amide alcohols	*cmlA*, *floR*, *catB4*	Chloramphenicol, florfenicol
Quinolones	*aac(6’)-Ib-cr*	Nalidixic acid, enrofloxacin, norfloxacin, dafloxacin
Tetracycline	*sul*I, *sul*II	Sulfisoxazole, sulfamethoxazole
Efflux pump membrane transport coding genes	*bepD*, *bepE*, *bepF*	
MDR protein-encoding genes	*mdtA*, *mdtB*, *mdtC*, *mdtE*, *mdtG*, *mdtH*, *mdtK*, *mdtL*, *marA*, *marB*, *marC*, *marR*, *emr*	
Fosfomycin resistance Protein-encoding gene	*fsr*	Fosfomycin
Acridine yellow resistance protein-encoding gene	*acrA*, *acrB*, *acre*, *acrF*	Acridine yellow
Macrolide efflux pump protein-encoding genes	*macA*, *macB*	Macrolide
Metal ion resistance protein encoding genes	*merA*, *merB*, *merC*, *merE*, *merP*,*merT*, *merR*, *TerA*, *TerB*, *TerC*, *TerD*, *pcoA*, *pcoB*, *pcoC*, *pcoD*, *TerE*, *TerW*, *TerX*, *TerY*, *TerZ*, *zraP*	Mercury, antimony, copper, zinc
Bleomycin resistance protein-encoding genes	*Ble, bcr*	Bleomycin
Quaternary ammonium salt-resistance protein-encoding genes	*qacF*, *sugE*	Quaternary ammonium compound

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
