# Peer review of "Genomic Sequence Analysis of the Multidrug-Resistance Region of Avian Salmonella enterica serovar Indiana Strain MHYL"

_microorganisms, 2019, doi:10.3390/microorganisms7080248_

Round 1
Reviewer 1 Report
This is a simple genetic sequence of an antibiotic resistant strain of Salmonella Indiana. the resistance genes are located on the chromosome. Sentence structure throughout could use some polishing, and this manuscript is highly overwritten. Some data does not seem to be useful.
LIne 82: change disrupted to digested.
Line 85-87. Poor description.
Figure one, is sloppily put together, but I think all they are trying to show is that all the resistance genes are genomically located - but they do entire genomics sequencing? So this figure is not required?? See Figures 2 and 3??? The entire PCR protocol and tables 1 and 2 therefore are also not needed?
Lines 148 - 152 is methods? Why is it in the results? Beside this other materials in the results belong in the methods section.
Lines 160-161. Did the use a S. Typhimurium strain for genome assembly? Is there any issue with using other Salmonella species genomes for the assembly of these genomes?
Table 3 like figure one is not needed.
The results and discussion basically repeat the same thing, and the results and discussion section should be combined and greatly reduced in size.
Author Response
Line 82: change disrupted to digested.
Response: Thank you, the suggested change was made. (Line 91)
Line 85-87. Poor description.
Response: Thank you, we have improved the description of this important step. (Lines 92–111)
Figure one, is sloppily put together, but I think all they are trying to show is that all the resistance genes are genomically located – but they do entire genomics sequencing? So this figure is not required?? See Figures 2 and 3??? The entire PCR protocol and tables 1 and 2 therefore are also not needed?
Response: Part 1: Thank you, but we feel that Figure 1 was not sloppily put together. It did need improvement in the use of English, and we made those corrections and re-made Figure 1. Figure 1 shows the results of hybridization of resistance genes on the MHYL genome. (Line 170)
Part 2: Thank you, but this paper is a scientific research article, not a briefing. PCR is a very good method to use to assess the resistance genes, but that in itself was not able to show if the resistance genes resided in one or two fragments of the genome. Therefore, it was also necessary to sequence the whole genome of strain MHYL. Therefore, based on our studies both PCR and whole genome sequencing was important to understand the whole picture and it is necessary to include Tables 1 and 2. (Lines 60–64)
Lines 148 – 152 is methods? Why is it in the results? Beside this other materials in the results belong in the methods section.
Response: Thank you, this section was moved to the Materials and Methods. (Lines 127–132) Other material was also moved to the M&M section. (Lines 125–126)

Reviewer 2 Report
The manuscript by Lu et.al, is interesting. They have mapped the MDR regions of avian Salmonella isolate and has nicely commented on the mechanism by which the isolate may have gained multi drug resistance. These kinds of studies are helpful in dealing with MDR issues in recent time. There are minor corrections which are to be taken care of:
1) Line 19: MDR genes carried on ...... Carried can be omitted
used to position genes........ Can be changed to- locate genes
2)Line 43-49: It will be good if the author can reconstruct the sentences.
3)Line 82 : DNA was then disrupted......Can be changed to then digested.
4)Line 119: The genomic DNA of ..........The extracted genomic DNA
5)Line 213: IS26 element or the like........The statement can be restructured.
6) Figure3 : proA-proR....need to be changed to pcoA-pcoR as mentioned in the manuscript
Author Response
1) Line 19: MDR genes carried on …… Carried can be omitted
Used to position genes……… Can be changed to- locate genes
Response: Thank you, the changes have been made. (Line 19)
2)Line 43-49: It will be good if the author can reconstruct the sentences.
Response: Thank you, the sentences were improved. (Lines 47–52)
3)Line 82 : DNA was then disrupted …… Can be changed to then digested.
Response: Thank you, the suggested change was made. (Line 91)
4)Line 119: The genomic DNA of …….. The extracted genomic DNA
Response: Thank you, the suggested change was made. (Line 158)
5)Line 213: IS26 element or the like……The statement can be restructured.
Response: Thank you, “or the like” was deleted. (Line 241)
6) Figure3 : proA-proR…..need to be changed to pcoA-pcoR as mentioned in the manuscript
Response: Thank you very much, we have re-made Figure 3 and made your suggested changes. Again, thank you for pointing this change out.

Round 2
Reviewer 1 Report
The authors are to be commended for their English corrections in the revised manuscript.
Lines 97 to 111, are not required, this should be greatly reduced - these are very common procedures that do not add to this study.
Figure 1 still is sloppy, and these results can be easily handled in the text - once again Fig. 1 is not helpful. Does this figure represent multiple gel pictures that were cut up to produce this figure?????
Lines 175-179. The authors only need to state the coverage of the genome.
table 3. mar, emr, acr and mdt genes are all intrinsic low level drug resistance genes, and these are found in all Salmonella I suspect. The MDR assignment is correct but misleading. I suspect some other genes are also like these, perhaps the genes should be filed under a different heading, to demonstrate that they doe not disseminate?
Lines 307-318 - seem to be results? some of this is speculative as well: Conducive to capturing? unconventional recombination? be transmitted horizonatally?
Author Response
Microorganisms-550525
Reviewer 1
1) Lines 97 to 111, are not required, this should be greatly reduced - these are very common procedures that do not add to this study.
Response: Thank you, the suggested change was made. (Line 97)
2) Figure 1 still is sloppy, and these results can be easily handled in the text – one again Fig. 1 is not helpful. Does this figure represent multiple gel pictures that were cut up to produce this figure?????
Response: Thank you. Figure 1 was deleted as you suggested. Section 3.1. was modified accordingly.
Lines 175-179. The authors only need to state the coverage of the genome.Response: Thank you. , the suggested change was made. (Line 158-160)
4) table 3, mar, emr, acr and mdt genes are all intrinsic low level drug resistance genes, and these are found in all Salmonella I suspect. The MDR assignment is correct but misleading. I suspect some other genes are also like these, perhaps the genes should be filed under a different heading, to demonstrate that they doe not disseminate?
Response: Thank you. We have changed the headings in Table 3. This should not be misleading now.
5) Lines 307-318 - seem to be results? some of this is speculative as well: Conducive to capturing? unconventional recombination? be transmitted horizonatally?
Response: Thank you, the suggested change was made. (Line 287) We removed the speculative part from the result.